# Computed Tomographic Epidurography in Patients with Low Back Pain and Leg Pain: A Single-Center Observational Study

**DOI:** 10.3390/diagnostics12051267

**Published:** 2022-05-19

**Authors:** Kimiaki Yokosuka, Kimiaki Sato, Kei Yamada, Tatsuhiro Yoshida, Takahiro Shimazaki, Shinji Morito, Kouta Nishida, Atsushi Matsuo, Takuma Fudo, Naoto Shiba

**Affiliations:** Department of Orthopedic Surgery, Kurume University School of Medicine, 67 Asahi-machi, Kurume-shi 830-0011, Japan; kimiaki@med.kurume-u.ac.jp (K.S.); yamada_kei@kurume-u.ac.jp (K.Y.); yoshida_tatsuhiro@kurume-u.ac.jp (T.Y.); shimazaki_takahiro@med.kurume-u.ac.jp (T.S.); morito_shinji@med.kurume-u.ac.jp (S.M.); nishida_kouta@med.kurume-u.ac.jp (K.N.); matsuo_atsushi@kurume-u.ac.jp (A.M.); fudou_takuma@med.kurume-u.ac.jp (T.F.); nshiba@med.kurume-u.ac.jp (N.S.)

**Keywords:** CT epidurography, epidural adhesiolysis, epidural connective tissue

## Abstract

This study was conducted to analyze the findings and benefits of computed tomography (CT) epidurography in patients with low back and leg pain and compare these findings with those of magnetic resonance imaging (MRI) images. In total, 495 intervertebral discs from 99 patients with low back and leg pain who underwent percutaneous epidural adhesiolysis (epidural neuroplasty or percutaneous adhesiolysis) were examined. The axial views of CT epidurography were classified into six types to examine each intervertebral disc: round type, ellipse type, spike type, Benz mark, incomplete block, complete block, and non-contrast. MRI images were graded from A to D using the Schizas classification. Notably, 176 images were round-type and ellipse-type axial views, and 138 were spike-type and Benz-mark views; Schizas classification Grades A and B were observed in 272 and 47 MRI images, respectively. The incomplete block and complete block axial images did not significantly differ in CT epidurography and Schizas classification Grades C and D. The images showing Benz marks existed only at the L4/5 and L5/S intervertebral levels and only in 14.7% of patients. The ratio of normal shadows differed between MRI images and CT epidurography. Therefore, CT epidurography may enable a detailed evaluation of the epidural space.

## 1. Introduction

The lumbar spinal canal stenosis causes low back and leg pain due to stenosis of the spinal canal. The stenotic spinal canal varies from the midline to the lateral recess in elevated and stenotic areas.

Due to numerous factors related to low back pain, and the low specificity of diagnostic imaging and clinical diagnosis, the diagnosis and treatment of low back pain and leg pain are often difficult. Clinically, MRI is the most used diagnostic tool and there are various treatments that may be used, including drug therapy, rehabilitation, and shockwave therapy [1,2].

The most effective diagnostic tools for lumbar spinal canal stenosis are magnetic resonance imaging (MRI) and myelography. These minimally invasive and convenient tools are widely applied because of their effectiveness in diagnosing most spinal disorders. However, Andrasinova et al. [3] found a significant correlation between the neurological impairment score in lumbar spinal canal stenosis and Schizas morphologic classification. Pain intensity, walking capacity, and functional disability displayed no correlation with the MRI parameters of lumbar spinal canal stenosis. Therefore, it remains difficult to diagnose epidural lesions using MRI. Epidural space lesions (adhesion) may be one of the causes of symptoms in lumbar spinal canal stenosis [4]. Additionally, percutaneous epidural adhesiolysis using a transsacral approach to the epidural space has significant effects on certain lesions, such as epidural space adhesion [5,6,7,8,9]. The spinal epidural space between the spinal dura mater and spinal canal is rich in venous plexus and adipose tissue. MRI is effective for visualizing lesions in the spinal canal but not in the so-called epidural space between the spinal canal and dura mater. The only test method for identifying epidural space is epidurography (peridurography). With the advent of MRI in 1978, epidurography gradually became obsolete. Currently, convenient myelography and MRI are primarily used in the clinic. MRI is the most useful diagnostic test for spinal disorders and can detect tumors in the spinal canal and spinal stenosis lesions [10]. However, MRI cannot detect the pathology of local inflammation and adhesion and cannot be used for dynamic evaluation, occasionally leading to errors in the definitive diagnosis of vertebral spinal injury levels. Epidurography is the most useful test for the static and dynamic imaging of epidural space lesions and adhesion.

Epidurography confirms the correct position of the epidural catheter and identifies the adhesion site of the epidural space, but also has a therapeutic effect, such as a runoff pattern contrast, in the treatment of epidural adhesiolysis (neuroplasty). It is also used as a guide to judge the correct position of the epidural catheter [11,12]. Evaluation of the degree of epidural adhesion using CT epidurography has not been reported. Seeling et al. observed the rail-track phenomenon or plica mediana dorsalis, but the findings are limited to anatomical references, with no mention of its frequency of appearance or pathology [13]. This method is useful for imaging patients with low back and leg pain when the MRI images show no abnormalities in the epidural space. Moreover, there are no reports comparing MRI and CT epidurography. 

We hypothesized that patients with low back and leg pain without abnormalities on MRI would show abnormalities on CT epidurography. This study was conducted to classify CT epidurography findings in patients with low back pain and leg pain and compare these findings with those obtained from MRI images.

## 2. Materials and Methods

### 2.1. Patients

This single-center, prospective, observational study was performed from August 2018 to April 2020, with approval from the institutional review board (approval number: 19139) at our center. Written informed consent was obtained from all patients involved in the study. 

A total of 495 intervertebral discs level from 99 patients with low back and leg pain who underwent percutaneous epidural adhesiolysis were examined. Patients who were treated for adhesiolysis and were able to undergo CT epidurography after treatment for adhesiolysis were included. Among the 99 patients, 51 were male and 48 were female, with a mean age of 72.7 years. The mean body mass index was 24.05 (Table 1). Senior experts orthopedic with spinal experience performed the diagnosis of low back pain and lumbar stenosis.

### 2.2. Clinical Assessment

For image evaluation, the axial views of CT epidurography were classified into the following six types to examine the intervertebral discs between the L1 and S1 vertebrae: round type, ellipse type, spike type, Benz mark, incomplete block, complete block, and non-contrast (Figure 1). In the axial CT images, if a normal shadow of the contrast medium that accumulates in the epidural space is uniformly depicted in a donut-like circle, it is considered the round type; however, in some cases, stenosis is not observed on MRI. Nevertheless, in the image, one edge of the shadow is incorrect. If the shadow of the contrast medium appears wider than the round type, it is known as the ellipse type. An image showing that the dura is pulled toward the ligamentum flavum because of the influence of epidural connective tissue is known as the spike type, which adheres more firmly to the dura. In the Benz-mark type, the contrast medium appears as a rhombus. A complete block is defined as a remarkable blockage of the contrast medium on the image; if this blockage is not remarkable, the result is considered an incomplete block. A non-contrast type is defined as when the contrast medium is indistinguishable. In addition, MRI findings obtained from these patients were compared with the CT epidurography images using the Schizas classification [14]. 

Unclear images of the intervertebral discs and those with low contrast sensitivity were classified as non-contrast. Imaging diagnosis using MRI and CT epidurography of the lumbar spine was performed by two senior experts in spinal disorders. When the two experts disagreed, the images were classified as non-contrast. To maintain the accuracy of the image, images that could be evaluated were classified as non-contrast. MRI and CT epidurography images were sorted by number and evaluated at different places and times. The MRI model used in this study was Discovery MR750w Expert3.0T (GE Healthcare, Little Chalfont, UK), and the CT model used was a Discovery CT750 HD scanner (GE Healthcare). Percutaneous epidural adhesiolysis included the fluoroscopic placement of an epidural catheter (myeloCath, Biomedica Healthcare, Inc., Tokyo, Japan) within the sacral epidural space under local anesthesia with the patient in the prone position. The medication (total of 20 cc) used in this study consisted of 9.5 mL saline + 0.5 mL dexamethasone sodium phosphate (decadron injection 1.65 mg; Sun Pharma Japan, Ltd., Tokyo, Japan) + 10 mL mepivacaine hydrochloride solution (carbocaine ampoule injection 1%; Nihon-pharm). The contrast agent (10 cc) was iotrolan (isovist injection 240; non-ionic, cistern, spinal cord, and articular; Bayer Yakuhin, Ltd., Tokyo, Japan). CT imaging was performed immediately after percutaneous epidural adhesiolysis. MRI can be used as a reference for CT epidurography performed after treating epidural connective tissue adhesion. All patients were initially diagnosed using MRI. CT epidurography was performed for anatomical evaluation of the epidural space. Percutaneous epidural adhesiolysis was performed by a specialist. The therapeutic inclusion criteria were patients with low back pain, radicular pain, or cauda equina symptoms, and without arteriosclerosis obliterans, regardless of the presence or absence of spinal stenosis on MRI of the lumbar spine, who underwent at least 4 weeks of appropriate conservative treatment (drug therapy and/or physical therapy). The exclusion criteria (because of high risk) were patients with severe spinal stenosis, progressive paralysis, bleeding tendency, ossification of the ligament, injury (e.g., fracture dislocation), or those who could not maintain a prone position. MRI was performed before initiating treatment for epidural adhesion, and CT epidurography was conducted immediately after epidural adhesion.

### 2.3. Statistical Analysis

A chi-square test was used to evaluate the correlation between the MRI and CT epidurography findings. A *p*-value of <0.05 was considered to indicate that the results were statistically significant.

## 3. Results

The spinal diseases evaluated in this study were lumbar spinal stenosis (including degenerative spondylolisthesis) (*n* = 80), lumbar disc herniation (*n* = 4), facet cysts (*n* = 1), severe degenerative scoliosis (*n* = 1), MRI of the lumbar spine with no obvious anomalies (unknown) (*n* = 13), and a history of surgery for the lumbar spine (*n* = 19, 19.2%) (Table 2).

To maintain inter-rater reliability, 122 non-contrast images, including images showing a discrepancy between the two experts, were excluded; the experts examined 373 images.

MRI images were classified as Grade A (*n* = 272), Grade B (*n* = 47), Grade C (*n* = 52), and Grade D (*n* = 2), using the Schizas classification (Table 3). CT epidurography types consisted of round type (*n* = 88), ellipse type (*n* = 88), spike type (*n* = 83), Benz mark (*n* = 55), incomplete block (*n* = 57), and complete block (*n* = 2) (Table 4). Images showing the Benz marks were observed only at the L4/5 and L5/S intervertebral levels in 14.7% (55/373) of patients (L4/5 level (*n* = 22) and L5/S1 level (*n* = 33)). 

Notably, 176 images were round and ellipse types, and 138 images were spike and Benz-mark types, according to CT epidurography. Overall, 272 and 47 images showed Grades A and B in Schizas classification. Therefore, the images obtained using CT peridurography and MRI significantly differed. There was no significant difference between the incomplete-block or complete-block axial views on CT epidurography and Schizas classification Grades C and D (Table 5). 

Furthermore, in cases in which spinal canal stenosis on MRI of patients with low back pain or leg pain could not be observed, epidural connective tissue may have affected the outcome of CT epidurography, resulting in a spike- and Benz-mark-type axial view.

Analysis of the CT epidurography type in Schizas classification suggested that Grades A and B on MRI with *p* < 0.001 showed a correlation to abnormal findings on CT (Table 6).

## 4. Discussion

### 4.1. CT Epidurography

CT epidurography is useful for diagnosing spinal canal stenosis and epidural space adhesions. Dural sacs that were imaged using CT epidurography show a homogeneously clear edge and clear visualization of the epidural space (Figure 2). In addition, the sensitivity of CT epidurography for diagnosing a typical lumbar spinal stenosis is equivalent to that of MRI (Figure 3). Furthermore, in the case of postoperative adhesion between the inner surface of the spinal canal and dura mater, CT epidurography can detect epidural space adhesion that cannot be detected using CT myelography (Figure 4). Therefore, CT epidurography is highly useful for diagnosing the extent and severity of adhesions. In addition, even in the absence of obvious stenosis lesions in the spinal canal, CT epidurography shows anomalies in the dural sac (Figure 5). The classification of CT epidurography (Figure 1) developed in this study is sorted in descending order based on the morphology of the dural sac and the ratio of the cross-sectional area of the dural sac to that of the spinal canal. Simultaneously, the classification for the axial view CT epidurography is sorted in ascending order based on the ratio of the cross-sectional area of the epidural space to that of the spinal canal. There is a difference in the ratio of Grade A in the Schizas classification and the ratio of round and ellipse types in CT epidurography; this difference is greater in the lower lumbar spine.

Adhesions may occur in the epidural space, even if MRI does not clearly show stenosis, which is particularly noticeable at the lower lumbar level and must be considered when some patients show no abnormalities on MRI and complain of low back, leg, and buttock pain.

Figure 1 shows a selection of common morphologies on CT epidurography. Epidural spaces that could not be classified were labeled as non-contrast. Morphology shows the state and adhesion of dorsomedial connective tissue (DCT). However, the relationship between morphology and symptoms is unknown. We evaluated epidural spaces with a Benz-mark-type axial view on epidural CT, which were concentrated at L4/5 and L5/S intervertebral levels. Studies are needed to examine the clinical relevance of these findings.

Round and ellipse types, spike type and Benz mark, complete block, and incomplete block were difficult to distinguish in the lower lumbar spine by the two reviewers, and images with unclear shadows were classified as non-contrast. Strong adhesion of the epidural space, particularly after spinal surgery, caused CT failure, making the result difficult to interpret.

### 4.2. Presence of Epidural Connective Tissue

Classification of images in this study improved the clarity of the morphological characteristics of epidural space. The round and ellipse types revealed the absence of spinal stenosis and epidural space narrowing, as well as the absence of adhesion to the connective tissue. The spike type showed a state of dura mater with traction to the connective tissue band and may also show a reduction of epidural space size in the spinal canal. Among all types, the Benz mark, which was concentrated in the lower lumbar spine, may show a significant reduction in the epidural space size based on the traction in three directions. This phenomenon strongly suggests that epidural lesions contribute to clinical symptoms. In addition, connective and fibrous tissues are important for surgery outcomes [15,16]. Future studies should explore methods for improving the detection of symptoms by processing the DCT, such as under optical observation with a fiber video scope.

Detailed analysis of the localization of lumbar disc herniation and epidurographic findings by Yamamura et al. showed that the predictive value of diagnosing lumbar disc herniation by epidurography was 91.3% [17]. Sasahara et al. showed that the findings of patients with leg pain and radicular symptoms who underwent epidurography were more consistent with epidurographic findings than with MRI findings and that MRI cannot adequately detect the pathology of local inflammation and adhesion [18]. When the pathology cannot be diagnosed using MRI, an inspection of the epidural space may be useful for diagnosing and treating patients with non-specific low back pain. In addition, many studies reported the presence and significance of DCT. Savolaine et al. [19] discussed the presence of posterior midline epidural connective tissue, such as plica mediana dorsalis [20], in the evaluation of CT epidurography in 40 patients. Shi et al. examined 30 donated tissues and reported the incidence of dorsal meningovertebral ligaments at the L5/S1 level as 97%, suggesting that dorsal meningovertebral ligaments cause injury to the dura matter and epidural hemorrhage during laminectomy [21]. Connor et al. examined the anatomical and histological characteristics of posterior epidural ligaments between the lumbar dural sac and yellow ligament in 17 donated tissues and detected posterior epidural ligaments in 52.9% of cadavers primarily at the L4/5 level [22]. Seeling et al. examined the findings of CT epidurography of the thoracic level in 30 patients to identify the appropriate location for epidural catheter placement [13]. They observed the absence of epidural space in the ventral part of the spinal canal at the thoracic spine level and determined the clinical importance of diagnostic tests for epidural space recognition. Blomberg [23] confirmed the presence of connective tissue between the posterior dural sac and yellow ligament in 48 patients who underwent epiduroscopy and identified (1) the dura mater with an extremely tight morphology of connective tissue that is maintained very close to the yellow ligament and (2) membrane formation in the dorsal sagittal plane when the dura adheres to the yellow ligament. Thus, various connective tissues may be elliptical because of excessive traction of the dural sac. Some connective tissues show various shadows on CT epidurography because of the posterior (dorsal) retraction caused by a tight morphology or duraplasty. However, the detailed mechanisms remain unclear. These anomalies in the epidural space may cause unknown intractable low back pain and leg pain. Therefore, determining the mechanisms of epidural space lesions may be useful for developing diagnosis and treatment methods in pain therapy.

### 4.3. Limitations

This study only involved image evaluation and did not consider clinical symptoms or therapeutic effects. Studies of the patient characteristics in clinical evaluation are needed. Additionally, the timing of MRI and CT epidurography differed, and the CT images may have been affected by percutaneous epidural adhesiolysis.

## 5. Conclusions

The difference in images between MRI and CT epidurography suggests that patients with low back pain and leg pain and without obvious spinal canal stenosis have abnormalities in the epidural space. Therefore, CT epidurography may allow detailed evaluation of the epidural space, which cannot be diagnosed using MRI. Currently, no definite relationship exists between the therapeutic effects of CT epidurography and its images. Our results suggest that DCT is involved in CT epidurography. However, it was difficult to determine the clinical significance of our results based on the present classification. Future studies are needed to examine the relationship of these imaging findings with clinical symptoms, therapeutic effects, and other factors to improve pain research.

## Figures and Tables

**Figure 1 diagnostics-12-01267-f001:**
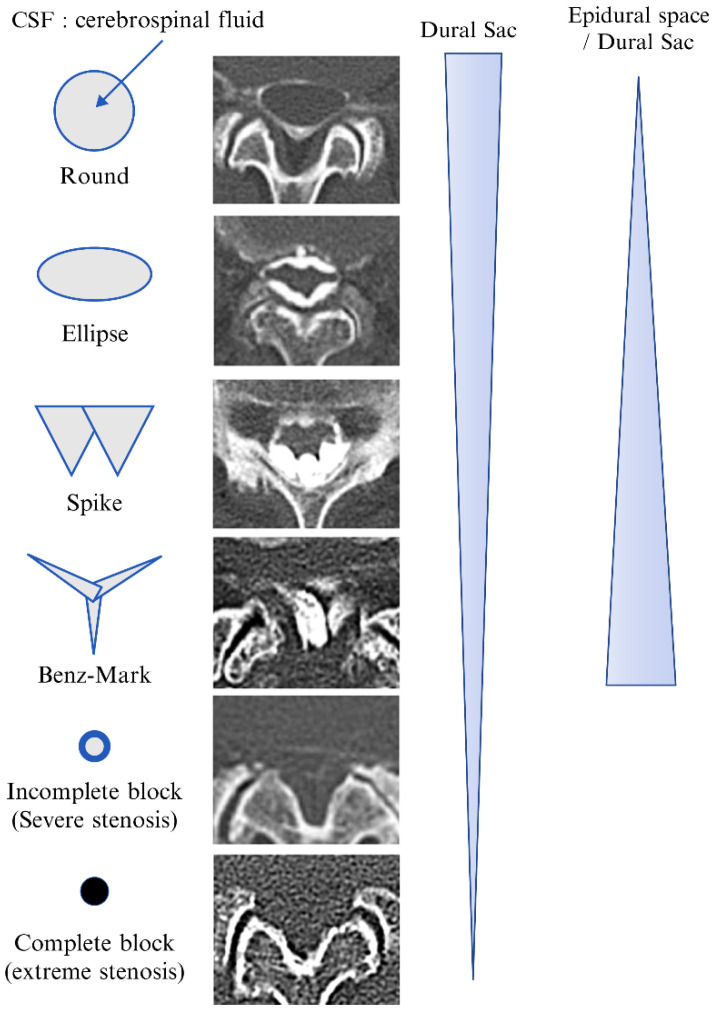
Classification of computed tomography epidurography. CSF, cerebrospinal fluid.

**Figure 2 diagnostics-12-01267-f002:**
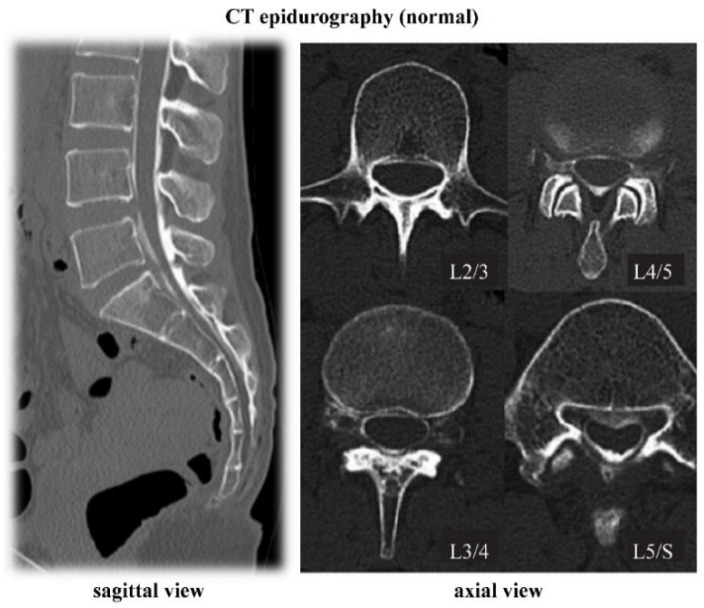
CT epidurography (normal). Sagittal view and axial view. CT, computed tomography.

**Figure 3 diagnostics-12-01267-f003:**
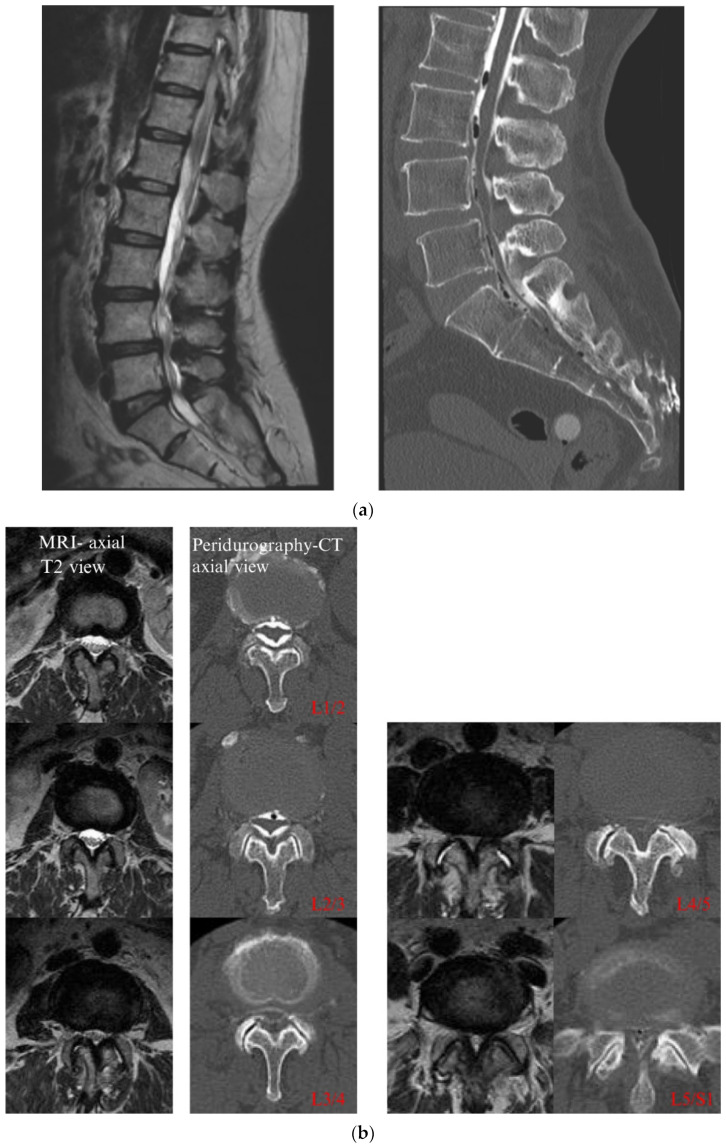
(**a**) Lumbar spinal canal stenosis. MRI sagittal T2 view and CT epidurography sagittal view. CT, computed tomography; MRI, magnetic resonance imaging. (**b**) Lumbar spinal canal stenosis. MRI axial T2 view and CT epidurography axial view. CT, computed tomography; MRI, magnetic resonance imaging.

**Figure 4 diagnostics-12-01267-f004:**
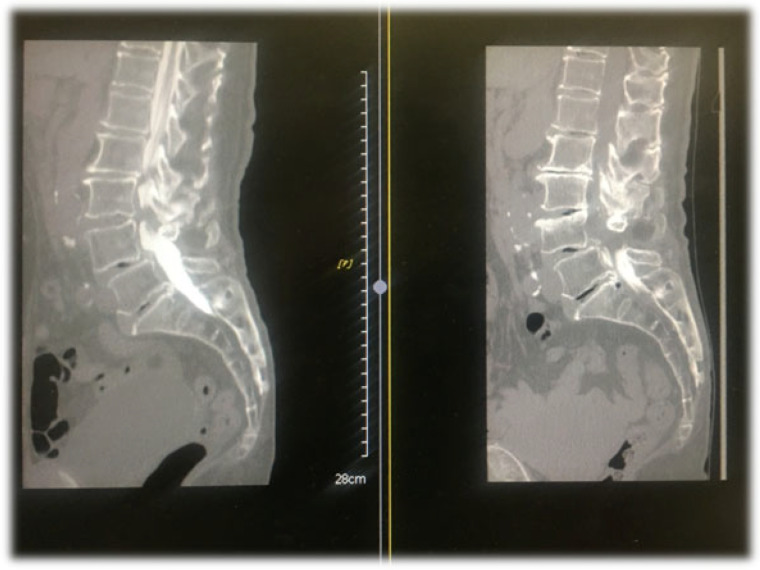
Epidural space adhesions. CT myelography sagittal view and CT epidurography sagittal view. CT, computed tomography; MRI, magnetic resonance imaging.

**Figure 5 diagnostics-12-01267-f005:**
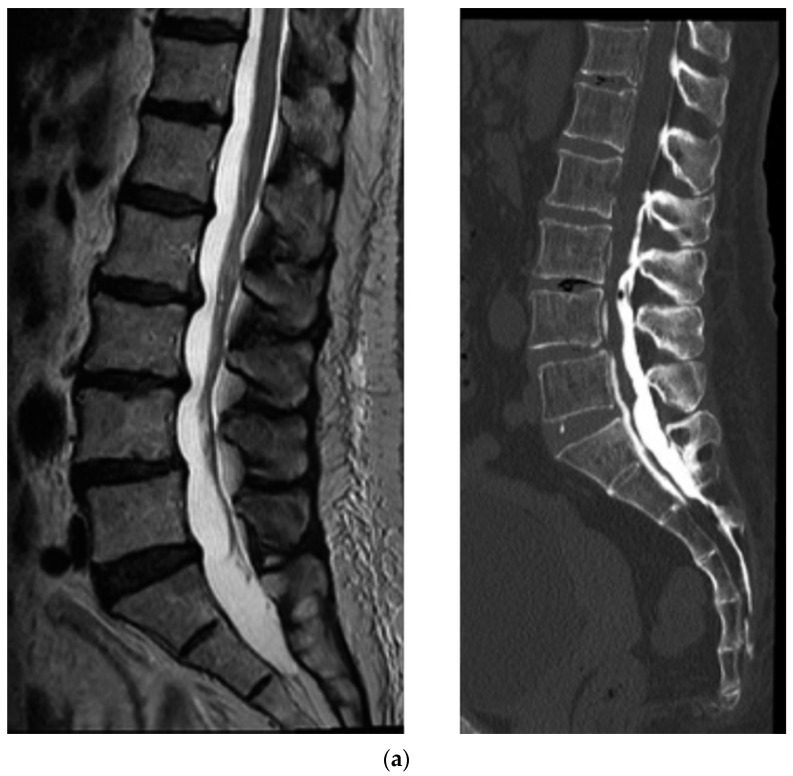
(**a**) MRI sagittal T2 view and CT epidurography sagittal view. CT, computed tomography; MRI, magnetic resonance imaging. (**b**) MRI axial T2 view and CT epidurography axial view. CT, computed tomography; MRI, magnetic resonance imaging.

**Table 1 diagnostics-12-01267-t001:** Patient demographic data.

No. of Patients	99
Age (years) (range)	72.7
Sex (M/F) (% male)	51/48 (51.5%)
Average height (cm)	157.3
Weight (kg)	60.2
Average BMI (range)	24.05
Percentage (%) of patients with obesity (BMI > 25%)	29.70%

BMI, body mass index.

**Table 2 diagnostics-12-01267-t002:** Diagnoses.

Diagnosis	*n*
Lumbar spinal canal stenosis (including degenerative spondylolisthesis)	80
Lumbar disc herniation	4
Facet joint cyst	1
Highly degenerate scoliosis	1
Cause unknown	13
Total	99

**Table 3 diagnostics-12-01267-t003:** Schizas classification (MRI).

	L1/2 (*n*)	L2/3 (*n*)	L3/4 (*n*)	L4/5 (*n*)	L5/S1 (*n*)	Total (*n*)
A1	48	35	19	26	56	184
A2	3	3	6	5	6	23
A3	5	9	7	5	2	28
A4	4	10	11	7	5	37
Total	60	57	43	43	69	272
B	4	8	17	13	5	47
C	6	5	13	22	6	52
D	0	0	0	1	1	2
Total (*n*)	70	70	73	79	81	373

MRI, magnetic resonance imaging.

**Table 4 diagnostics-12-01267-t004:** Computed tomography epidurography.

	L1/2 (*n*)	L2/3 (*n*)	L3/4 (*n*)	L4/5 (*n*)	L5/S1 (*n*)	Total (*n*)
Round	40	26	13	4	5	88
Ellipse	16	23	25	14	10	88
Spike	10	15	20	19	19	83
Benz mark	0	0	0	22	33	55
Incomplete block	4	6	15	18	14	57
Complete block	0	0	0	2	0	2
Total	70	70	73	79	81	373
Non-contrast	29	29	26	20	18	122

**Table 5 diagnostics-12-01267-t005:** Comparison of MRI and CT Epidurography.

CT Epidurography		Schizas Classification	
Round and ellipse	176	A	272
Spike and Benz mark	138	B	47
Incomplete block	57	C	52
Complete block	2	D	2
Total	373	Total	373
Non-contrast	122	Non-contrast	122

**Table 6 diagnostics-12-01267-t006:** **CT** Epidurography type in Schizas classification Grades A and B (expected value *p* < 0.001).

Type	A (*n*)	B (*n*)	Total (*n*)
Round and ellipse	171	5	176
Spike and Benz mark	101	37	138
Incomplete block	0	5	5
Complete block	0	0	0
Total	272	47	319
Expected value			
Type	A	B	
Round and ellipse	150.1	25.9	
Spike and Benz mark	64.1	21.1	
Incomplete block	4.26	0.74	
Complete block	0	0	

## Data Availability

The data presented in this study are available on request from the corresponding author. The data are not publicly available at the direction of the ethics committee.

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
