# Peer review of "Computed Tomographic Epidurography in Patients with Low Back Pain and Leg Pain: A Single-Center Observational Study"

_diagnostics, 2022, doi:10.3390/diagnostics12051267_

Round 1

Reviewer 1 Report

Dear Authors the topic is very interesting.

As regards the introduction i suggest to improve the introduction speaking about the causes of low back pain. The Author could describe the epidemiological impact of this disease, symptomatology link to disc degeneration and spinal stenosis. For this brief description i suggest to cite the following article in which Authors reported these aspects .

“Extracorporeal shockwave therapy versus exercise program in patients with low back pain: Short-term results of a randomised controlled trial

Journal of Biological Regulators and Homeostatic AgentsVolume 32, Issue 2, Pages 385 - 3891 March 2018

Notarnicola A . et al.”

As regards M&M and inclusion criteria, the Authors have to specify who performed the diagnosis of low back pain, lumbar stenosis. In fact it is important to report if there were an involment of neurosurgeon or orthopedic with a spinal experience in order to have an homogenous sample.

As regards the discussioni s supported by results and the sections are well balanced.

The conclusioni s very important because the Authors underlined the importance of CT epidurography for diagnostic iter of lumbar stenosis

Author Response

Dear Reviewer:

We/I wish to re-submit the manuscript titled “Computed tomographic epidurography in patients with low back pain and leg pain: A single-center observational study.” The manuscript ID is diagnostics-1726257.

We thank you for your thoughtful suggestions and insights. The manuscript has benefited from these insightful suggestions. I look forward to working with you to move this manuscript closer to publication in Diagnostics.

The manuscript has been rechecked and the necessary changes have been made in accordance with the reviewers’ suggestions, with changes in the manuscript marked with red font. The responses to all comments have been prepared and attached herewith.

Thank you for your consideration. 

Response to Reviewer 1 Comments

Point 1: As regards the introduction i suggest to improve the introduction speaking about the causes of low back pain. The Author could describe the epidemiological impact of this disease, symptomatology link to disc degeneration and spinal stenosis. For this brief description i suggest to cite the following article in which Authors reported these aspects .

“Extracorporeal shockwave therapy versus exercise program in patients with low back pain: Short-term results of a randomised controlled trial, Journal of Biological Regulators and Homeostatic AgentsVolume 32, Issue 2, Pages 385 - 3891 March 2018

Notarnicola A . et al.”

Response 1: Thank you for this suggestion. I added a sentence citing the proposed paper to the Introduction.

Point 2: As regards M&M and inclusion criteria, the Authors have to specify who performed the diagnosis of low back pain, lumbar stenosis. In fact it is important to report if there were an involment of neurosurgeon or orthopedic with a spinal experience in order to have an homogenous sample.

Response 2:  A sentence clarifying this aspect was added to Materials and Methods, 2.1. Patients. 

Reviewer 2 Report

I read with great interest the study: " Computed tomographic epidurography in patients with low back pain and leg pain: A single-center observational study". 

The idea of the authors is interesting andthe study si overall well conducted. 

Scientific method is clear. 

English quality is good. 

Conclusions are in line with obtained results. 

Discussion is clear and synthetic. 

As stated in the limitation section, the the weakest point is complete lack of clinical correlations of your imaging finding with pain and neurological symptoms of included patients. This is the main limit of the study.

A part that I would just suggest to expand your bibliographic research adding more references, here there are soma you can include for example:

  • Avellanal M. Epiduroscopia [Epiduroscopy]. Rev Esp Anestesiol Reanim. 2011 Aug-Sep;58(7):426-33. Spanish. doi: 10.1016/s0034-9356(11)70107-4. PMID: 22046865.
  • Gillespie G, MacKenzie P. Epiduroscopy--a review. Scott Med J. 2004 Aug;49(3):79-81. doi: 10.1177/003693300404900301. PMID: 15462218.
  • Igarashi T, Hirabayashi Y, Seo N, Saitoh K, Fukuda H, Suzuki H. Lysis of adhesions and epidural injection of steroid/local anaesthetic during epiduroscopy potentially alleviate low back and leg pain in elderly patients with lumbar spinal stenosis. Br J Anaesth. 2004 Aug;93(2):181-7. doi: 10.1093/bja/aeh201. Epub 2004 Jun 11. PMID: 15194631.
  • Kawanishi M, Kawase H, Kumagaya K. [Equipment for epiduroscopy and its clinical applications]. Masui. 2006 Sep;55(9):1112-7. Japanese. PMID: 16984009.
  • Colosimo C, Gaudino S, Alexandre AM. Imaging in degenerative spine pathology. Acta Neurochir Suppl. 2011;108:9-15. doi: 10.1007/978-3-211-99370-5_3. PMID: 21107932.
  • Avellanal M, Diaz-Reganon G, Orts A, Gonzalez-Montero L, Riquelme I. Transforaminal Epiduroscopy in Patients with Failed Back Surgery Syndrome. Pain Physician. 2019 Jan;22(1):89-95. PMID: 30700072.
  • Avellanal M, Diaz-Reganon G. Interlaminar approach for epiduroscopy in patients with failed back surgery syndrome. Br J Anaesth. 2008 Aug;101(2):244-9. doi: 10.1093/bja/aen165. Epub 2008 Jun 13. PMID: 18552347.
  • Yıldırım HU, Akbas M. Percutaneous and Endoscopic Adhesiolysis. Agri. 2021 Jul;33(3):129-141. English. doi: 10.14744/agri.2020.70037. PMID: 34318919.
  • Bellini M, Barbieri M. A comparison of non-endoscopic and endoscopic adhesiolysis of epidural fibrosis. Anaesthesiol Intensive Ther. 2016;48(4):266-271. doi: 10.5603/AIT.a2016.0035. Epub 2016 Sep 6. PMID: 27595746.
  • Helm S 2nd, Racz GB, Gerdesmeyer L, Justiz R, Hayek SM, Kaplan ED, El Terany MA, Knezevic NN. Percutaneous and Endoscopic Adhesiolysis in Managing Low Back and Lower Extremity Pain: A Systematic Review and Meta-analysis. Pain Physician. 2016 Feb;19(2):E245-82. PMID: 26815254. 

Author Response

Dear Reviewer:

We/I wish to re-submit the manuscript titled “Computed tomographic epidurography in patients with low back pain and leg pain: A single-center observational study.” The manuscript ID is diagnostics-1726257.

We thank you for your thoughtful suggestions and insights. The manuscript has benefited from these insightful suggestions. I look forward to working with you to move this manuscript closer to publication in Diagnostics.

The manuscript has been rechecked and the necessary changes have been made in accordance with the reviewers’ suggestions, with changes in the manuscript marked with red font. The responses to all comments have been prepared and attached herewith.

Thank you for your consideration. 

Response to Reviewer 2 Comments

Point 1: A part that I would just suggest to expand your bibliographic research adding more references, here there are soma you can include for example:

Response 1: Thank you for the suggested literature. From the proposed papers, I selected 5 references for the introduction and added them to the reference list.
